# Estimating population immunity against serotype-two poliomyelitis from the inactivated polio vaccine in routine immunization across 112 countries: A modelling study

Elizabeth J. Gray[1], Laura V. Cooper[1], Alejandro Ramirez Gonzalez[2], Ondrej Mach[3], Nieves Derqui[1], Nicholas C. Grassly[1], Isobel M. Blake[1]*

1 MRC Centre for Global Infectious Disease Analysis, School of Public Health, Imperial College London, London, United Kingdom, 2 Expanded Programme on Immunization, Vaccines, and Biologicals Department, World Health Organization, Geneva, Switzerland, 3 Polio Eradication Department, World Health Organization, Geneva, Switzerland

* isobel.blake@imperial.ac.uk

## Abstract

### Background

To mitigate the risk of outbreaks of serotype 2 poliomyelitis after withdrawal of this serotype from oral poliovirus vaccine (OPV) in 2016, inactivated poliovirus vaccine (IPV) was introduced into the routine immunization (RI) programmes of all countries using OPV. Since 2022, WHO has recommended a 2-dose schedule, with a first dose at 14 weeks of age followed by a second dose at least 4 months later (e.g., 14–39 week schedule), although an earlier schedule may be adopted, despite lower immunogenicity, if vaccine coverage is low at older ages.

### Methods and findings

We combined published data on type-2 IPV seroconversion with age, national RI coverage estimates, dose introduction dates, and country-specific schedules using a cohort model of population immunity to estimate IPV-induced immunity from 2024–2031 for 112 countries using either one or two doses of IPV. We projected immunity for current, 6–14, and 14–39 week schedules to find the optimal schedule and estimate the impact of interventions such as schedule changes and catch-ups. Under current schedules, estimated median serotype 2 population immunity in 2025 among children under five years of age is at 61% (IQR: 52%, 72%), rising to 71% (IQR: 57%, 80%) in 2031. The later 14–39 week schedule was optimal in all countries, with potential for the median immunity to rise to 78% (IQR: 66%, 85%) by 2031 if adopted by all countries in 2026. Eight countries would still have <50% immunity, rising to 65%−72% if catch-up campaigns with 80% coverage were implemented in 2030. The work is limited by the fact that IPV provides only a

which permits unrestricted use, distribution, and reproduction in any medium, provided the original author and source are credited.

**Data availability statement:** Custom code used in this study is available at https://github.com/ElizGray/IPV_induced_immunity_in_112_OPV_using_countries. Data on vaccination schedules can be found in this repository (compiled originally from https://immunizationdata.who.int/). A summary of seroconversion study data can be found in the 'Seroconversion_studies_table.csv' file of the GitHub repository. A download of this repository can be found at https://doi.org/10.5281/zenodo.18375210.

**Funding:** I.M.B received funding from the World Health Organization (Grant No. 2024/1500224-0): Funder website: www.who.int. Additionally, N.C.G. and I.M.B acknowledge funding from the MRC Centre for Global Infectious Disease Analysis (reference MR/X020258/1), funded by the UK Medical Research Council (MRC). This UK funded award is carried out in the frame of the Global Health EDCTP3 Joint Undertaking. Funder website: www.ukri.org/councils/mrc/. The funders had no role in study design, data collection and analysis, decision to publish, or preparation of the manuscript.

**Competing interests:** I have read the journal's policy and the authors of this manuscript have the following competing interests: A.R.G; O.M. are employees of the World Health Organization and contributed to the manuscript during the editing and data compilation stages. O.M. is the Team Lead for polio research at the World Health Organization. A.R.G. is a Technical Officer at the World Health Organization.

**Abbreviations:** bOPV, bivalent oral polio vaccine; IPV, inactivated poliovirus vaccine; OPV, oral poliovirus vaccine; RI, routine immunization; SAGE, Strategic Advisory Group of Experts on Immunization; tOPV, trivalent oral polio vaccine; VDPV, vaccine-derived poliovirus; VDPV2, serotype-2 VDPV; VAPP, vaccine-associated paralytic polio; WUENIC, WHO and UNICEF estimates of national immunization coverage

partial picture of total immunity where there has been emergency type-2 OPV use. Furthermore, national estimates may mask subnational coverage differences and pockets of extremely low immunity.

## Conclusions

Under these estimates, IPV schedules and coverage are suboptimal in many countries. Those with a single dose should introduce a second on the 14–39 week schedule; those on early schedules would benefit from adopting the 14–39 week schedule. IPV catch-up campaigns are recommended where RI coverage is low.

---

## Author summary

### Why was this study done?

- Multiple vaccine-derived type-2 poliovirus outbreaks continue to circulate.

- The inactivated polio vaccine (IPV) is the only vaccine delivered through routine immunization that protects against type-2 poliomyelitis. A minimum of two IPV doses is recommended to be given at 14 weeks of age and 4 months later, (e.g., 39 weeks), although 43 countries or territories administer an earlier week schedule, often at 6/14 weeks.

- There is a need to quantify type-2 population immunity from IPV given differences in country schedules, national coverage and the timing of the second dose introduction.

### What did the researchers do and find?

- We developed a model to estimate national population immunity against type-2 poliomyelitis across 112 countries in children <5 years old.

- We projected immunity levels for current immunization schedules, alternative scenarios, and catch-up vaccination campaigns.

- Current IPV-induced immunity levels are critically low in many locations, with 25 countries or territories having <50% population immunity in children under five years of age.

- We found that the IPV schedules and coverage are suboptimal in many countries, and in many cases adopting the 14–39 week schedule would be beneficial.

### What do these findings mean?

- The 43 countries or territories with an early immunization schedule, and the 23 countries using a single dose of IPV immunization are recommended to change to a 14–39 week schedule.

- Catch-up campaigns could be considered to shorten the time to reach higher immunity in countries with recent introductions of a second dose of IPV, and in countries where IPV coverage is very low and immunity would otherwise remain critically low.

- The work is limited by the fact that IPV provides only a partial picture of total immunity where there has been emergency type-2 live vaccine use. Furthermore, national estimates may mask subnational coverage differences and pockets of extremely low immunity.

## 1 Introduction

The inactivated polio vaccine (IPV) is effective in protecting against poliomyelitis of all three serotypes of poliovirus and is essential to mitigate the risk of paralysis during the polio endgame. In contrast to the live oral polio vaccine (OPV), it cannot be transmitted between individuals to cause outbreaks of vaccine-derived poliovirus (VDPV) [1].

Serotype-2 was globally removed from OPV in April 2016 in order to stop new outbreaks of serotype-2 VDPV (VDPV2) and vaccine-associated paralytic polio (VAPP) following the eradication of wild serotype two poliovirus (bivalent oral polio vaccine (bOPV) replaced trivalent oral polio vaccine (tOPV)). At least one dose of IPV in routine immunization was recommended by the Strategic Advisory Group of Experts on Immunization (SAGE) to be administered at least from the date of serotype-2 withdrawal [2]. This was to be introduced in each bOPV-using country to mitigate paralysis risk from continued VDPV2 transmission, or failure to contain the virus from immunodeficient persistent shedders [3], given the fall in protection against serotype-2 poliomyelitis from OPV. Since the tOPV-bOPV switch, there has been a vicious cycle of serotype-2 VDPV outbreaks and responses, with some low-quality OPV campaigns reintroducing type-2 vaccine virus into now low-immunity populations, leading to many children being paralysed [4]. Between May 2016 and August 2024 87 emergences have been reported [5]. Cases of paralytic circulating VDPV numbered 1072 in 2020 alone [6]. Despite some increased immunity in selected locations due to outbreak response activities, which is temporary as children age out of the under-five cohort, immunity from IPV in routine immunization now represents the only long-term routinely-provided protection against type-2 poliomyelitis. If future cases of paralysis are to be prevented, it is vital that protection against symptomatic infection in the under-five population be increased and remain high.

As of 2022, SAGE has recommended that IPV should be administered to children at 14 weeks, with a second dose more than four months later (suggested to coincide with other vaccines at 39 weeks). Countries may choose an "early schedule" of 6 and 14 weeks based on local epidemiology and where vaccination coverage at older ages may be low [6].

Many countries opt for a different schedule altogether, notably 9 and 17 weeks in many American-region countries. Due to global shortages of IPV following the tOPV-bOPV-switch, six countries switched to administering fractional IPV following evidence from clinical trials showing that two fractional doses of IPV, each one-fifth of a full dose [7], have a protective effect as least as good as a single full dose [8, 9].

IPV seroconversion rates improve with the age of administration [10] while routine immunization (RI) coverage tends to decline, as children drop out of their immunization schedule [7, 11, 12]. This decline varies and it is unclear which schedule would be optimal for each country. Furthermore, certain countries experience delays in vaccination, meaning children receive vaccination when they are older than the official scheduled age [13]. These delays in vaccination may affect the optimal schedule.

We estimate the current and future impact of IPV administered through RI in 112 bOPV-using countries which also use one or two doses of IPV and identify strategies to increase population immunity. IPV-induced immunity against type-2 poliomyelitis in the under-five population in each country is estimated, with projections to 2031, using WHO and UNICEF estimates of national immunization coverage (WUENIC) 2019–2023 [7], published schedules [7], and an updated review of seroconversion rates with age. Effects of various interventions are assessed, namely estimating whether a switch to

the early (6–14 week) or preferred (14–39 week) schedule in 2025 would improve immunity, by modelling the progression of immunity between the present day and 2031, under each assumption. The impact of introducing a second dose where countries have not yet done so is predicted, as are the effects of potential IPV catch-up campaigns for under-vaccinate children, following optimal schedule adoption. Finally, data from Demographic and Health Surveys [13] are used to estimate how predictions change accounting for country-specific RI administration delays.

## 2  Methods

### 2.1  Serotype-2 seroconversion estimates by age of administration

A systematic review of seroconversion against serotype-2 poliovirus by age of IPV dose administration for one or two full or fractional IPV doses [10] was updated in February 2023 using the same search terms given in the original publication. We then estimated the overall association of serotype-2 seroconversion as a function of age of administration by fitting a binomial regression model to these study data with linear and polynomial functions (as selected by the Akaike Information Criterion) of age at first dose and the length of delay between doses, accounting for study period (pre- or post tOPV-bOPV switch). Full details are in S1 Text Sect 4.2.

### 2.2  National routine immunization coverage estimates by age of vaccine administration

To predict the impact of different IPV schedules, RI coverage across all vaccines was characterised for each country as a function of age of vaccine administration. WUENIC coverage estimates from the five years preceding estimates (2019–2023) were obtained and matched to the ages of administration from official published schedules [7]. A linear model used age as a predictor of the logit function of the coverage estimate, enabling the estimation of likely coverage under alternative schedules.

### 2.3  Constructing IPV induced population immunity estimates against type-two poliomyelitis in children under 5 years old

We estimate national population immunity against type-2 poliomyelitis in children under five years of age in 112 bOPV-using countries administering one or two doses of IPV. N.B. we exclude Curaçao and Montserrat from aggregated estimates due to insufficient data.

For each point in time between January 2024 and January 2031, we estimate for each country or territory the proportion of children under the age of five years immune to type-2 poliovirus by calculating, for each birthdate in the preceding five years the probability that a child has received one or two doses of IPV based on age eligibility, historic schedules and coverage at scheduled ages, before multiplying this by the seroconversion rate for the relevant age of administration. This calculation integrates any changes to immunization schedules or coverage during that window, with coverage levels held constant at 2023 values thereafter. These daily probabilities are then averaged, assuming a uniform age distribution and birth rate to give the overall immunity proportion. Population immunity reaches equilibrium five years after any new dose introduction or schedule change. A more technical explanation is available in S1 Text Sect 4.3.

### 2.4  Assessing the impact of interventions

The impact of interventions on population immunity are estimated, specifically changes to either the 6–14 or 14–39 week schedule, and of subsequent catch-up IPV campaigns. It is assumed that schedule changes would apply to children born after 1st January 2026, and catch-up campaigns would take place in January 2027 or January 2030, giving single doses to 80% or 50% of children under five years of age who had missed one or both vaccinations through prior ineligibility or missed doses. Details can be found in S1 Text Sect 4.4.

## 2.5 Incorporating delay in IPV administration

Delays in vaccine administration can increase seroconversion probability, as IPV immunogenicity improves with age, while longer unvaccinated periods reduce overall immunity. The effect of these delays on the equilibrium immunity for different schedules is assessed using information on country specific delays in vaccine administration from recent Demographic Health Survey (DHS) [14] data, from which distributions of delay in receipt are constructed, as in [13, 15]. We consider delays for DTP1 (Diphtheria, Tetanus, and Pertussis first dose), DTP3 (Diphtheria, Tetanus, and Pertussis third dose), and MCV1 (Measles containing vaccine first dose) vaccines, given at the same ages as SAGE-recommended IPV schedules. Countries with surveys from 2013 onwards with >1,000 children were included. Detailed methods are in S1 Text Sect 4.5.

# 3 Results

## 3.1 Current IPV schedules

As of June 2025,134 countries or territories administer bOPV. Of these, 23 administer a single full dose of IPV, 86 two full doses of IPV, and 5 two fractional doses of IPV. India administers three fractional doses (Fig 1). The remainder administer three or more full doses. Seven countries use the early 6–14 schedule, 28 a 14–39 week schedule, 43 countries or territories give a first of two doses earlier than 14 weeks. Haiti and Curaçao give single doses at six and nine weeks, respectively. Of the 91 administering two doses, 44 introduced their second dose at the start of 2022 or later. Here it will be several years before the immunity of children under five years old reaches an 'equilibrium' position.

## 3.2 IPV induced type-two poliovirus seroconversion rates with age

Using seroconversion data from 19 studies, with 44 different arms, (26 full-dose, 18 fractional, with median 170 and 157 participants respectively) of which 22 (15 full dose, 7 fractional) were conducted before the 2016 tOPV-bOPV switch, we find that the rate of type-two seroconversion increases with both age at first dose and with the gap between doses for both fractional and full doses (Fig 2 and Table 1). Details of the studies considered may be found in S1 Text Sect 4.2.

We estimate relatively low seroconversion (28.3% 95% CI 23.6%, 33.6%) from a full single dose at 6 weeks of age, increasing to 68.7% (95% CI 64.1%, 73.0%) at 14 weeks of age. The seroconversion rate after a single full dose given at 14 weeks is less than 8% lower than after a two-dose 6–14 week schedule.

## 3.3 Declines in vaccination coverage by age

RI coverage declines with age in most (83/112) countries (Fig 3), (Fig B in S1 Text). In 19 countries, (17 in Africa), more than 10 % of children drop out of their immunization schedules between 14 and 39 weeks of age, most starkly in: Benin (75% to 48%), Democratic Republic of Congo (64% to 39%), and Chad (77% to 54%). At 14 weeks of age 97/112 countries have estimated typical coverage greater than 70%, falling to 82/112 countries at 39 weeks.

For countries in which there is no clear decline in coverage with age over the range of vaccines given (n = 29), we assume constant coverage at the mean coverage of available doses. Estimates for all countries and territories may be found in S1 Text Sect 1.

## 3.4 IPV induced population immunity against type-2 poliomyelitis

Assuming current schedules are maintained, IPV-induced immunity is forecast to increase in 92/112 countries between 2025 and 2031 (Fig 4), largely due to second-dose introductions. In January 2025 the country-wise median type-2 IPV induced population immunity among children under five years of age was predicted to be 61% (IQR:52%, 72%; median 54% and 63% in one and two dose countries respectively), rising by January 2026% to 64%, (IQR:53%, 75%; median 54%, 67%, in one and two dose countries and territories respectively), and to 71% (IQR:57%,80%, median: 54%, 74% in one and two dose countries or territories respectively). Adjusting for population size [16] this corresponds to immunity in

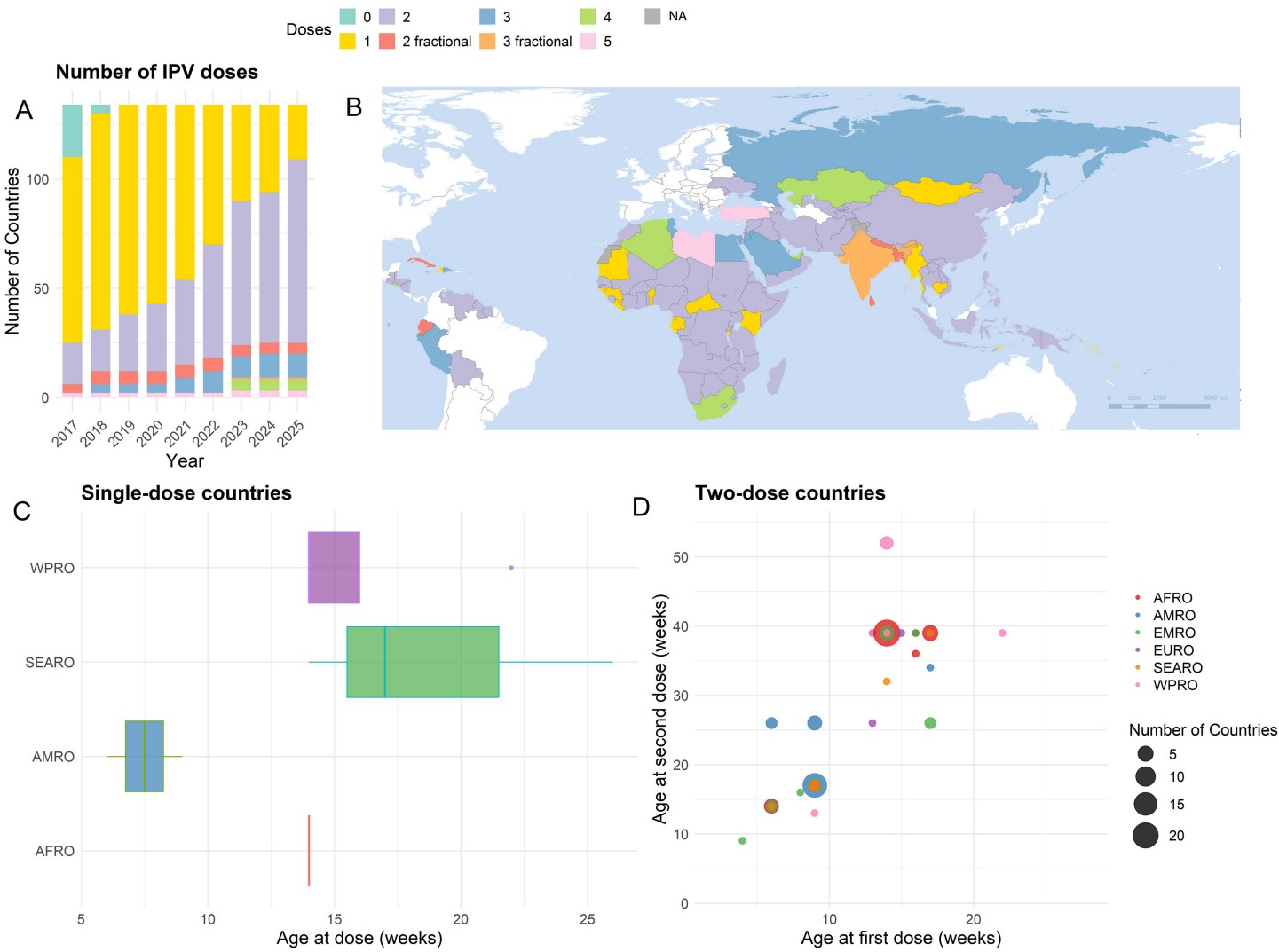

**Fig 1. Global schedule variation in Inactivated Polio Vaccine (IPV) administration through routine immunization for countries using the oral polio vaccine. A**: Number of current bivalent Oral polio Vaccine (bOPV)-using countries administering each number of IPV doses per year (data underlying Fig 1A can be found in S1 Text Sect 6, Table Y), **B**: Number of IPV doses administered by country in 2025, white denotes IPV-only countries; **C, D**: age of country scheduled administration, by WHO region, for one and two dose countries respectively. For the box plot, median, interquartile range, and range are shown. Moldova, administering doses at 26 and 104 weeks, has been omitted from the lower right plot. AFRO, African Region, AMRO, Region of Americas, EMRO, Eastern Mediterranean Region, EURO, European Region, SEARO, South-East Asia Region, WPRO, Western Pacific Region. NB. The boundaries and names shown and the designations used on this map do not imply the expression of any opinion whatsoever on the part of WHO concerning the legal status of any country, territory, city or area or its authorities, or concerning the delimitation of its frontiers or boundaries. The base map data were provided by the World Health Organization POLIS Geodatabase [17], and is publicly available from https://gis-who.hub.arcgis.com/pages/whogeodatabase.

61% of children living in the locations in question in 2026 (50.5%, 62% in one and two dose countries or territories respectively), progressing to 65% in 2030, (51%, 66% in one and two dose countries and territories respectively), dominated by China, giving doses at 9 and 13 weeks with an estimated immunity of 67%. African and Eastern Mediterranean countries tend to have the lowest immunity: Fig 4. 16 countries or territories are predicted to have <50% immunity by 2031, eight of

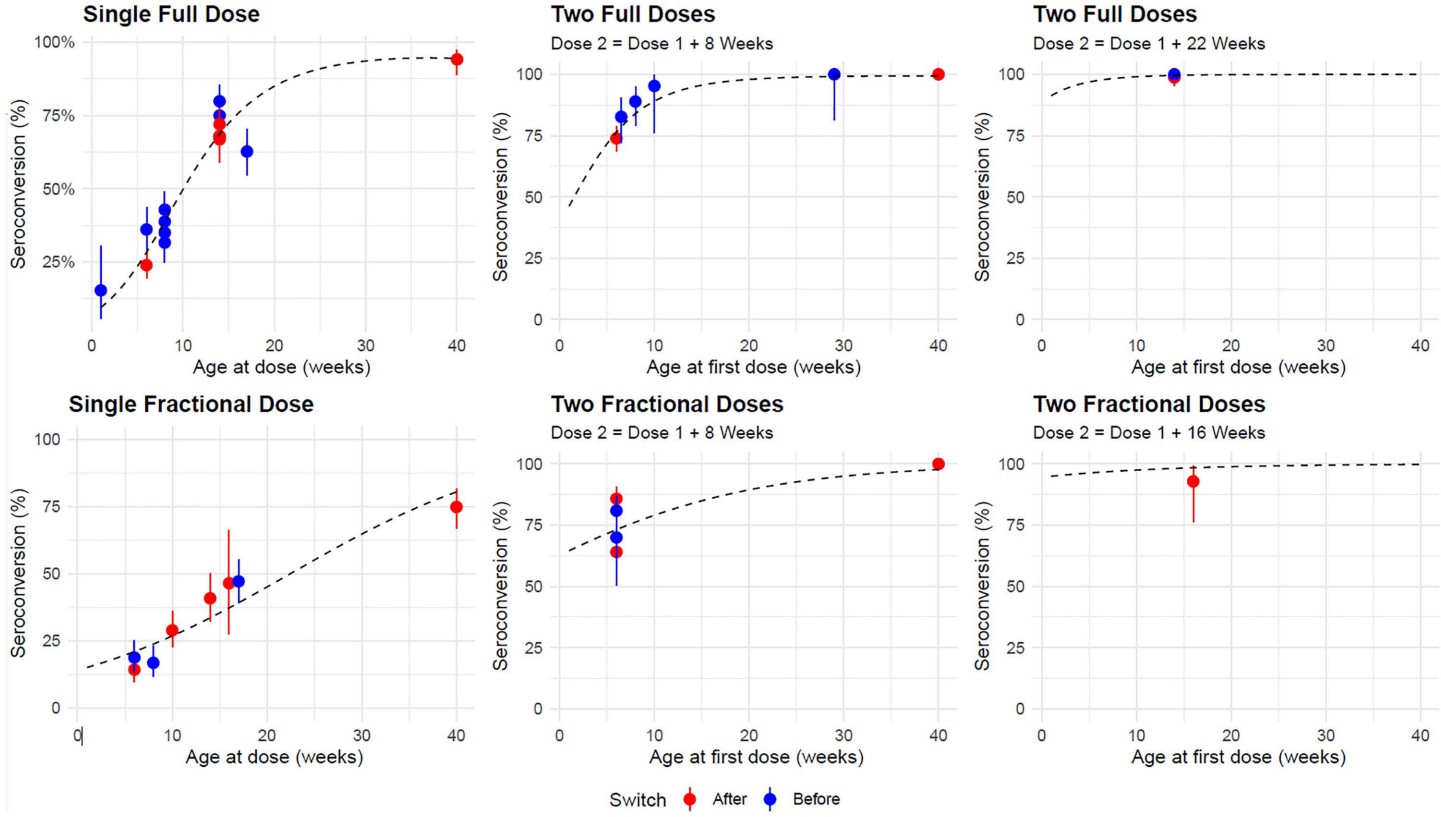

**Fig 2. Serotype 2 poliovirus seroconversion by age of administration for full (Upper) and fractional (Lower) doses (N studies = 26 and 18 respectively).** NB The three horizontal panels are snapshots of the same two-dimensional curve, for different values of the gap between dose one and two. As each point represents the seroconversion estimate from a different study with its own sample size, the error bars show the 95% confidence interval round the seroconversion estimate for that particular study.

**Table 1. IPV induced type-two polio seroconversion associated with different ages of administration and dosing, obtained through fitting a binomial regression model to a review of 19 seroconversion studies conducted between 1985 and 2022.**

| Schedule | Seroconversion rate after first dose (95% CI) | | Seroconversion rate after both doses (95% CI) | |
|---|---|---|---|---|
| 6, 14 weeks (full dose) | 28.3% | (23.6%, 33.6%) | 76.5% | (68.3%, 83.0%) |
| 14, 39 weeks (full dose) | 68.7% | (64.1%, 73.0%) | 99.7% | (98.9%, 99.9%) |
| 6, 14 weeks (fractional dose) | 20.9% | (12.7%, 32.6%) | 73.2% | (60.9%, 82.7%) |
| 14, 39 weeks (fractional dose) | 33.5% | (23.4%, 45.5%) | 99.9% | (99.0%, 99.98%) |

which have two doses (Angola, Papua New Guinea, Somalia, Yemen, Nigeria, Sudan, Occupied Palestinian Territories, Venezuela).

**3.4.1 Estimating the impact of the second dose of IPV and alternative schedules.** For all 112 countries or territories, a 14–39 week schedule results in higher immunity than a 6–14 week schedule (Fig 5). If all countries considered were to adopt the 14–39 week schedule in 2026, the projected country-wise median immunity would rise to 78% by 2031 (IQR: 66%, 85%), from a projected 71% (IQR: 57%, 80%) if current schedules stay the

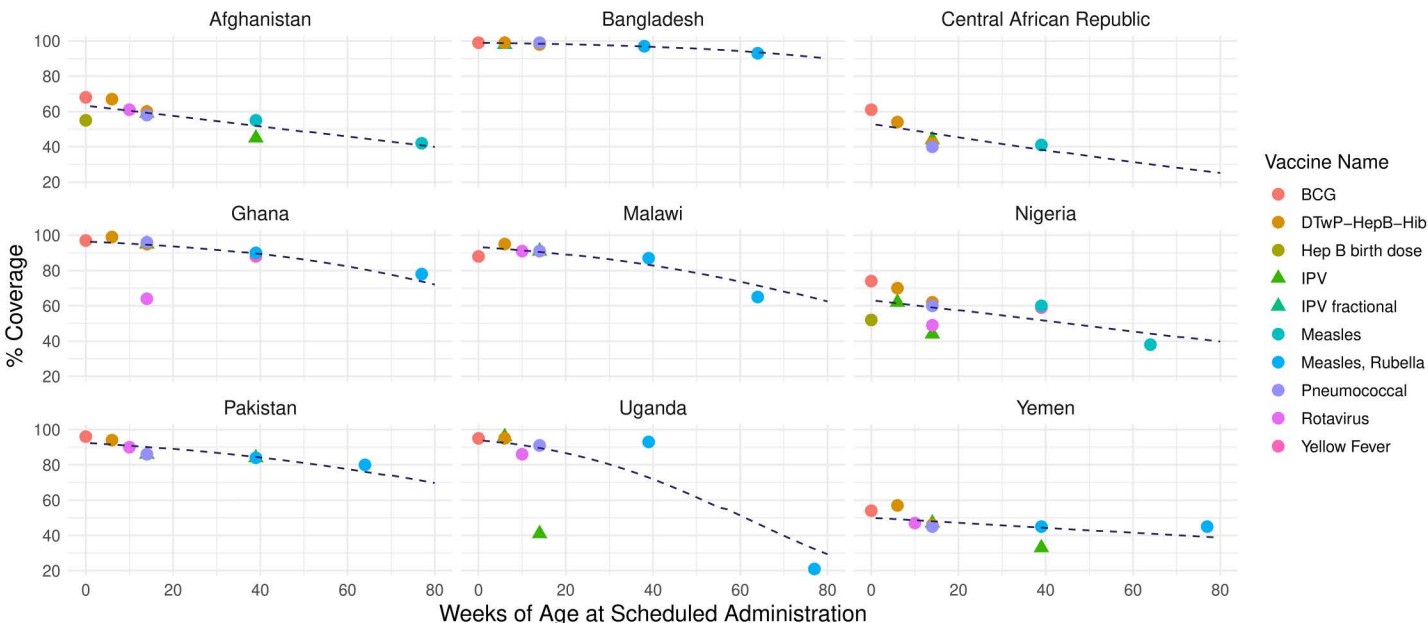

**Fig 3. Routine immunization coverage as a function of age of administration in 9 countries (points), with estimated coverage curves (lines).** Coverage estimates are obtained from 2023 WHO-UNICEF estimated national immunization coverage (WUENIC) data and country vaccination schedules from [7]. Vaccine name abbreviations: BCG, Bacillus Calmette–Guérin, DTwP-HepB-Hib = diphtheria, tetanus, whooping cough, hepatitis B and Haemophilus influenzae type B, Hep B = Hepatitis B, IPV, Inactivated polio vaccine.

same. Of 16 countries and territories predicted to have less than 50% immunity by 2031, five (Nigeria, Sudan, Occupied Palestinian Territories, Djibouti, and Venezuela) could increase immunity by 4%–38% by switching to the recommended schedule. Six countries with immunity below 50% administering single doses could improve immunity by 10%–36% by introducing a second dose (Fig 5). In particular, Haiti, administering a single dose at 6 weeks, could increase immunity from 18% to 54% by 2030 by adopting the 14–39 week schedule in 2026. Guinea and the Central African Republic could increase immunity from 36% to 47%, and 31% to 41%, respectively, by administering a second dose at 39 weeks.

China, currently using a nine, 13-week schedule, could increase immunity from 67% to 90% by switching to the standard schedule. Uganda and Nigeria, using a 6–14 week schedule, could increase immunity from 67% to 81% and 44% to 52%, respectively, by switching to the standard schedule. Estimates for all countries and territories that could improve immunity by schedule change are in in the S1 Text Sect 3.

Eight countries are predicted to have immunity below 50% by 2031, even if they have, or were to, introduce the standard schedule, due to low RI coverage across all ages: Central African Republic (41%), Guinea (47%), Sudan (48%), Venezuela (49%), Angola (45%), Papua New Guinea (34%), Somalia (39%), and Yemen (44%).

**3.4.2 Impact of catch-up campaigns.** The impact of catch-up campaigns following a switch to the 14–39 week schedule is shown in Fig 6 for 12 countries or territories. Where RI coverage is poor, the catch-up has a large initial impact, followed by a five-year decline to an equilibrium. In countries or territories with good coverage in 2023, but the earlier schedule, e.g., Uganda, Occupied Palestinian Territories, no decline is seen, as it is balanced by gains from the better schedule. Tables for further countries and territories are found in S1 Text Sect 2.

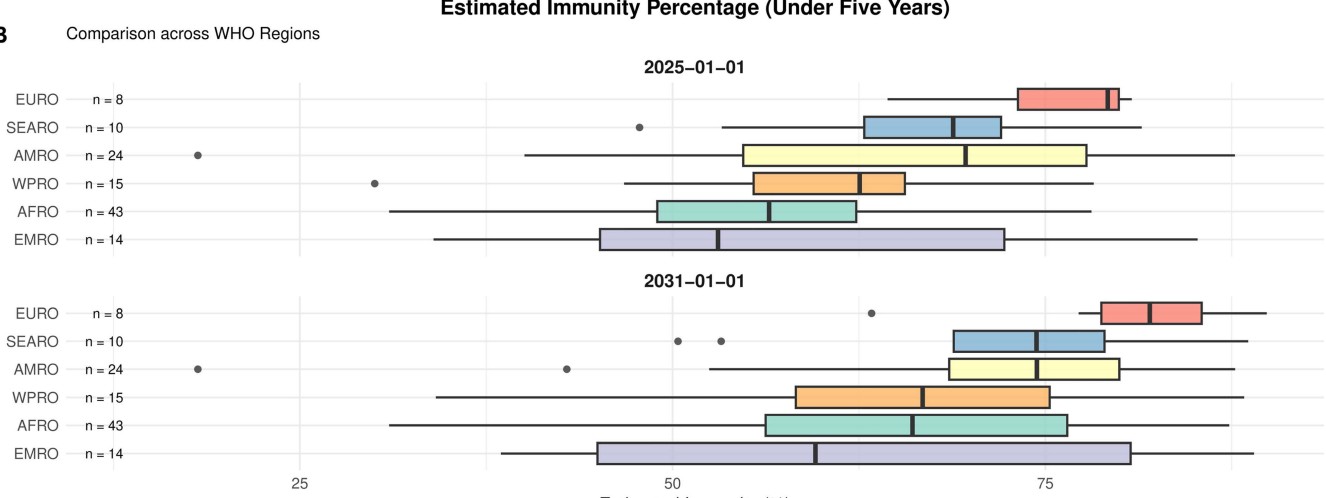

**Fig 4. Top to bottom: Estimated national IPV-induced population immunity against serotype-2 poliomyelitis for 112 bivalent oral polio vaccine (bOPV)-using countries and territories currently administering one and two doses of IPV in routine immunization, in January 2025, and January 2031.** By region, estimated immunity with current schedules, 2025, 2031, For the box plot, median, interquartile range, and range are shown.

WHO, World Health Organization, AFRO, African Region, AMRO, Region of Americas, EMRO, Eastern Mediterranean Region, EURO, European Region, SEARO, South-East Asia Region, WPRO, Western Pacific Region. NB. The boundaries and names shown and the designations used on this map do not imply the expression of any opinion whatsoever on the part of WHO concerning the legal status of any country, territory, city or area or its authorities, or concerning the delimitation of its frontiers or boundaries. The base map data were provided by the World Health Organization POLIS Geodatabase [17], and is publicly available from https://gis-who.hub.arcgis.com/pages/whogeodatabase.

### 3.5 Impact of delay in vaccine administration

Demographic Health Surveys with more than a thousand children were available for 36 countries or territories (median 5,959 children surveyed per country) (see S1 Text, Sect 4.5). More children received doses late, rather than early: for DTP1 the most common median delay among countries and territories was two weeks. In Nigeria, 20% of children surveyed who had received at least one dose of DTP received it early, while 67% received it late. Haiti and Chad have the greatest delays, whereby children report receiving DTP3 a median eight weeks after the scheduled 14 weeks. In contrast, median age of receipt of DTP1 and MCV1 by children from Maldives and Guinea is on time.

When including country-specific delays into the IPV immunity projections, 24/36 countries or territories are predicted to have greater immunity in children under 5 years old by 2031 than when doses are assumed to be on time, notably Haiti (31% instead of 18%), and Bangladesh (82% instead of 69%). Immunity in Nigeria is projected to be 49%, accounting for median delays of two and three weeks at six and 14 weeks of age, instead of 44% with no delay considered. Among the remaining countries for which immunity is projected to be lower when delay is considered, the difference is < 5%.

When vaccination delay is accounted for, the difference in outcomes from the early and standard schedule is lessened. Three countries have slightly higher immunity on the early schedule compared with the standard schedule when delay is assumed: Chad (67% versus 64%), Democratic Republic of Congo (53% versus 55%), Haiti (54% versus 55%). Full results in the S1 Text, Fig A.

## 4 Discussion

There is wide geographic variation in IPV-induced population immunity against poliomyelitis. Although we project a modest improvement in many countries and territories, driven by recent introductions of a second dose of IPV, IPV-induced immunity will remain low, projected to be < 50% by 2031 in 16 countries and territories. This is especially troubling given the persistent, multiple-continent circulation of type-two vaccine-derived poliovirus [17]. The variation is attributable to differences in vaccination coverage, choice of RI schedule and the timing of any schedule changes, and age-related immunogenicity.

Progress has been made in recent years, with 44 countries and territories introducing a second dose of IPV since the start of 2022, leaving only 23 still only administering a single dose. However, it is crucial to recognise that introducing a second dose without consideration of vaccination schedule is insufficient to raise population immunity, even with good coverage. Countries and territories introducing a second IPV dose on the early schedule see little improvement over a single dose, leaving children needlessly vulnerable to paralysis, with large programmatic costs for minimal gains in immunity. Countries and territories administering one dose should introduce a second, especially those with recent circulating VDPV2s, such as Benin, the Central African Republic, and Kenya. For countries and territories yet to implement the second dose, schedule choice is paramount. Countries and territories using the early schedule would benefit from switching to the preferred later 14–39 week schedule or similar. While switches are not straightforward, they are necessary: a first dose at 6 weeks induces immunity in less than a third of recipients, and two early-schedule doses give only slightly more immunity than a single dose at 14 weeks. More than 10% of children under five years old in seven countries or territories (Bangladesh, China, Djibouti, Occupied Palestinian Territories, Senegal, Sri Lanka, and Uganda) are unprotected against

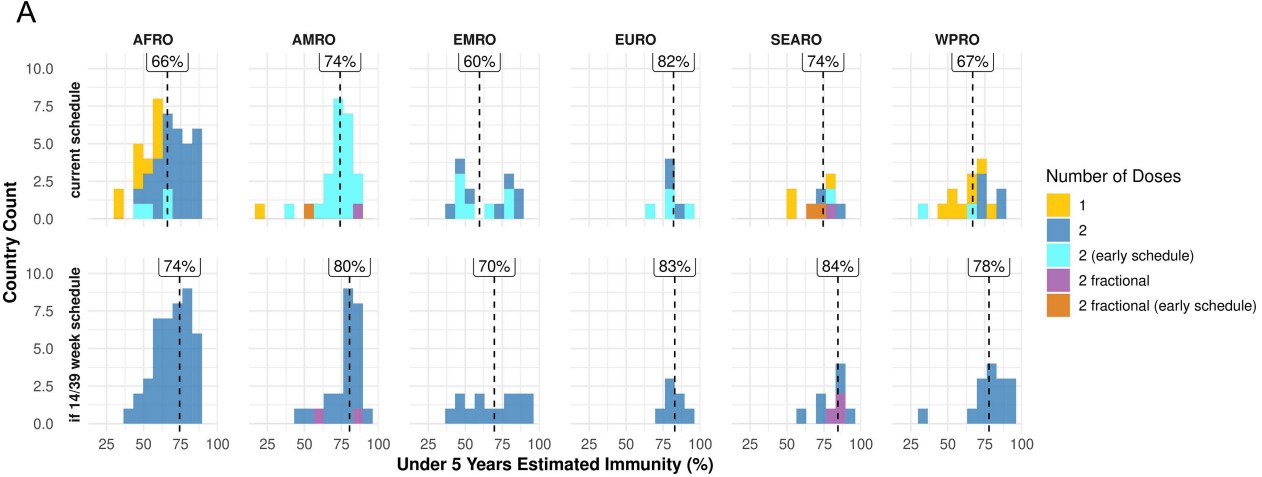

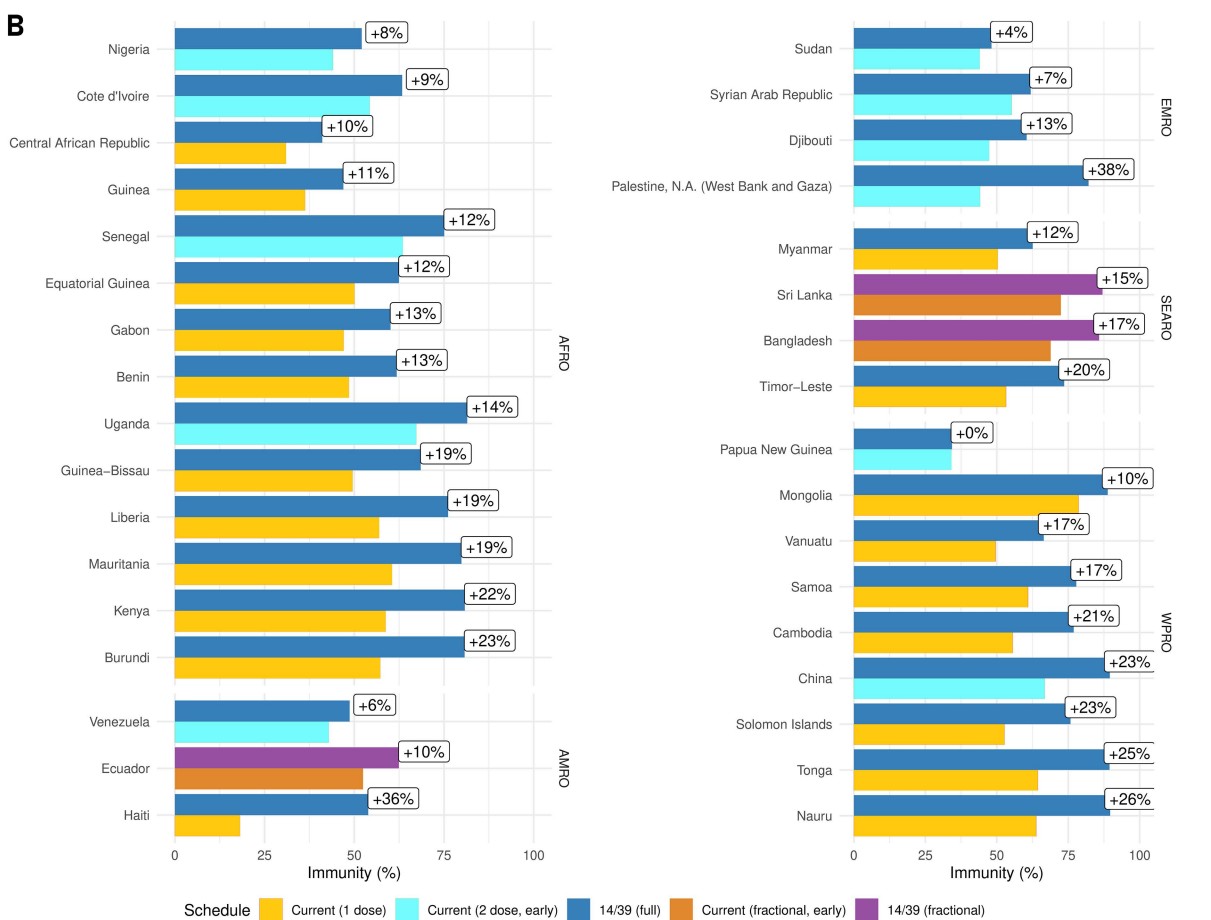

**Fig 5. Projected population immunity from inactivated polio vaccine (IPV) in routine immunization across different schedules for 112 bivalent oral polio vaccine (bOPV) using countries and territories A: Histogram of projected IPV induced type-two population immunity across 112 countries and territories, subdivided by region assuming the current schedule is maintained, and if all schedules are switched to a two-dose**

**14, 39 week schedule at the start of 2026.** Labels indicate median value. An early schedule is any with the first dose given before 14 weeks. Data underlying Fig 5A can be found in S1 Text Sect 2. B: Possible increases from schedule changes which could increase immunity by more than 10 percent of children under five, or in locations which otherwise have immunity in less than 60 % of children under five, for fractional and full (i.e., not fractional) doses. AFRO, African Region, AMRO, Region of Americas, EMRO, Eastern Mediterranean Region, EURO, European Region, SEARO, South-East Asia Region, WPRO, Western Pacific Region.

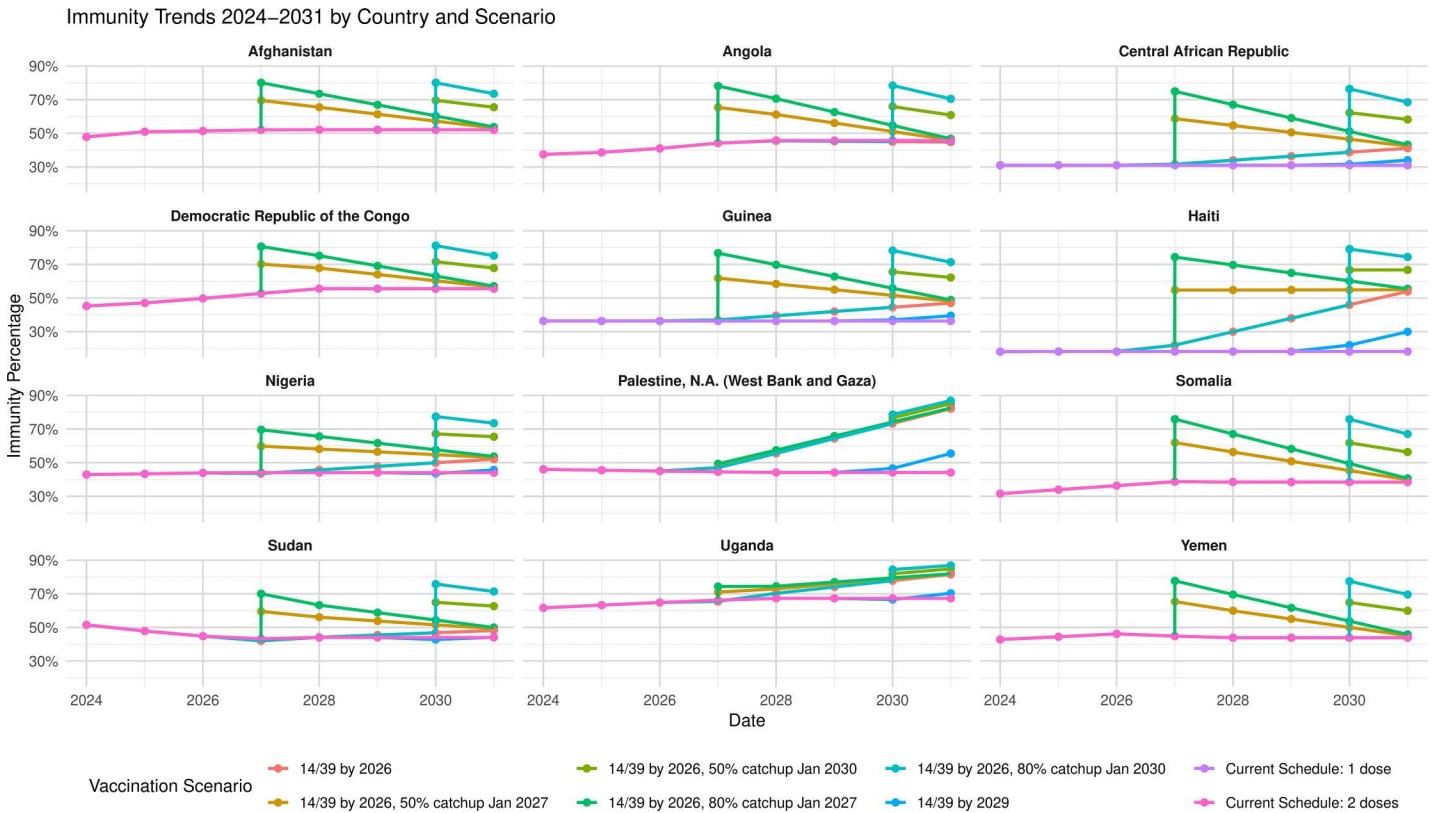

**Fig 6. Effects of possible interventions to increase IPV population immunity: changes of IPV schedule and catch-up campaigns with coverage of 50% or 80% of under-vaccinated children <5 years old.** Palestine, N.A. = Palestine National Authority.

paralysis who otherwise would be protected under the preferred schedule. Most countries and territories administer other vaccines at 39 weeks: schedule change would not require an additional health centre visit. Precedence exists in the case of Nepal, which will have increased its under five projected immunity from 64% to 79% by 2031, having switched in 2023 from the early to the preferred schedule, despite having only small improvements in coverage.

When doses are delayed, the difference between schedules is reduced. However, ethical questions surround leaving the children of those who adhere rigorously to the schedule less likely to receive protection. Additionally, constant review of the data would be necessary to ensure delays are maintained, keeping the early schedule optimal.

Sub-optimal early schedules lead to confusing policy messaging: in August 2024 following the paralysis of a child with polio in Gaza, a WHO release highlighted IPV coverage of >99% falling to <90% in the first quarter of 2024 [18]. These high coverages obscure a troubling immunity landscape and the severity of the situation: the Occupied Palestinian Territories have the least immunogenic two-dose IPV schedule of all bOPV-using countries and territories (four, eight weeks), leading to an estimated immunity of 46%, before any fall in coverage as a result of conflict is considered.

As of 2026, it is estimated that 21 and 76 of the 112 countries and territories would have immunity levels below 50% and 70% respectively. While transitioning all countries and territories to a two-dose schedule at 14–39 weeks would reduce this to 8 and 35 countries and territories respectively, achieving such levels would take until 2031 if schedules switched immediately. Catch-up campaigns could accelerate improvement, which is particularly important where there have been recent polio outbreaks. Additionally, campaigns would be beneficial in countries and territories with poor RI coverage already administering two doses to close immunity gaps, including regions at high risk of circulating VDPV2s. Although more expensive than oral polio vaccine campaigns, periodic targeted IPV catch-up campaigns in select locations could yield substantial benefit if higher coverage is achieved than through RI.

Looking to planned cessation of RI with all OPVs [19] and decline in mucosal immunity to poliovirus types one and three, high protection rates against symptomatic infection in the under-five population are vital. IPV second dose introduction and RI coverage will be a trigger for implementing bOPV cessation, however, risk assessments must consider country schedules and coverage when assessing risk mitigation from IPV. It is estimated that immunity to type-two poliovirus from IPV is comparable to immunity to types one and three [10]. In areas with low RI coverage, catch-up campaigns would be useful for preventing paralysis by these serotypes but should be timed appropriately to maintain high IPV-induced immunity following bOPV cessation.

Since December 2023, eligible countries and territories may apply for Gavi support in introducing the six-in-one Hexavalent vaccine [20]. The IPV component has a seroconversion rate of close to 100% when the recommended three-dose schedule is implemented [21], with a booster necessary at least six months after the third dose when the earliest 6, 10, 14 week schedule is adopted [6]. Some countries may intend not to introduce a second dose of IPV, instead waiting for this vaccine [22]. This is inadvisable, as it may be several years before Hexavalent introduction [23], leaving whole cohorts of children under-vaccinated and vulnerable to paralysis. The modelling approach we have developed to project IPV immunity may be used to predict the impact of the Hexavalent vaccine.

This work has several limitations. Firstly, IPV-induced immunity only provides part of the total immunity. To investigate overall vulnerability to paralysis, future work could consider serotype-2 oral polio vaccine doses given through campaigns in countries and territories experiencing VDPV2 outbreaks. Our estimates ignore immunity from maternal antibodies in the first six months of life, naturally acquired immunity, or protective effects from immune priming following a single IPV dose [24]. While a single IPV dose primes for a subsequent dose, indicating presence of immune memory, seroconversion and presence of neutralising antibodies at a titre >1:8 in blood are necessary for protection against poliomyelitis [10]. We have not included IPV campaigns or catch-ups, such as the fIPV campaign in Somalia in 2021 [25] or the 'Big Catch-Up' planned to target under-immunized children 2019–2022 [26], however these important events will not affect our 2030 immunity projections.

We have not given uncertainty estimates for population immunity because many of the data points lack uncertainty estimates (e.g., WUENIC, for RI coverage), and thus any derived uncertainty estimates which assume these to be known constants may be misleading. We have assumed that RI coverage remains constant from 2023 and while in general there are unlikely to be large fluctuations, there will be some exceptions where populations experiencing conflict (e.g., Occupied Palestinian Territories) experience declines in coverage.

When assessing the impact of schedule change, we have not considered that the number of injections a child is to be given at each clinic visit may affect coverage [27]. Future research should address this in collaboration with local health-care providers. When considering coverage, WUENIC are informed by multiple data sources and expert opinion. They are more reliable than administrative data but are not a perfect data source, nor, being whole-country estimates, can they capture subnational heterogeneity, which may be an important driver of polio risk where pockets of very low coverage exist [28]. In such places smaller scale targeted interventions may be more appropriate or cost-effective. This is particularly pertinent where areas have greater poliovirus exposure. Likewise, there is the possibility that using national estimates of coverage may cause inaccurate projections if localised confounding factors have a substantial effect on both coverage and the rate at which children drop out of their immunization schedule.

Future work could consider validation of the approach used in this study using the results of seroprevalence surveys, e.g., [29–31]. This has not yet been feasible as many prior surveys are limited in relevance due to their timing around tOPV cessation or OPV2 outbreak responses, being carried out in the context of now obsolete IPV schedules, or may include sampling bias from requiring at-clinic presentation in otherwise low coverage settings IPV is crucial for mitigating risks in the final stages of polio eradication by closing the humoral immunity gap and preventing paralysis. Benefits depend on schedule, coverage, and introduction time. Our findings emphasise the importance of selecting the recommended later schedule. While immunity gaps persist, catch-up campaigns would prove invaluable for GPEI, especially where VDPV2 circulates and in preparation for bOPV cessation.

## Supporting information

**S1 Text. Supporting information: This file includes both supplementary methods and results.**
(PDF)

## Acknowledgments

The authors alone are responsible for the views expressed in this article and they do not necessarily represent the views, decisions or policies of the institutions with which they are affiliated. The publication of the maps in this manuscript does not imply the expression of any opinion whatsoever on the part of WHO concerning the legal status of any territory, city, or area or of its authorities, or concerning the delimitation of its frontiers or boundaries.

## Author contributions

**Conceptualization:** Elizabeth J. Gray, Laura V. Cooper, Nieves Derqui, Nicholas C. Grassly, Isobel M. Blake.

**Data curation:** Elizabeth J. Gray, Alejandro Ramirez Gonzalez, Isobel M. Blake.

**Formal analysis:** Elizabeth J. Gray.

**Investigation:** Elizabeth J. Gray.

**Methodology:** Elizabeth J. Gray, Nieves Derqui, Isobel M. Blake.

**Writing – original draft:** Elizabeth J. Gray.

**Writing – review & editing:** Laura V. Cooper, Alejandro Ramirez Gonzalez, Ondrej Mach, Nieves Derqui, Nicholas C. Grassly, Isobel M. Blake.

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
