## [Editor Report · Decision Letter 0]

27 Feb 2025

Dear Dr Gray,

Thank you for submitting your manuscript entitled "Estimating population immunity against serotype-two poliomyelitis from the inactivated polio vaccine in routine immunisation across 114 countries" for consideration by PLOS Medicine.

Your manuscript has now been evaluated by the PLOS Medicine editorial staff as well as by an academic editor with relevant expertise and I am writing to let you know that we would like to send your submission out for external peer review.

Please re-submit your manuscript within two working days, i.e. by Mar 03 2025 11:59PM.

Kind regards,

Suzanne De Bruijn, PhD

Senior Editor

PLOS Medicine

---

## [Decision Letter · Decision Letter 1]

29 May 2025

Dear Dr Gray,

Many thanks for submitting your manuscript "Estimating population immunity against serotype-two poliomyelitis from the inactivated polio vaccine in routine immunisation across 114 countries" (PMEDICINE-D-25-00561R1) to PLOS Medicine. The paper has been reviewed by four subject experts and a statistician; their comments are included below and can also be accessed here: [LINK]

As you will see, the reviewers found the study interesting and well done. However there were a few methodological considerations raised that should be addressed. After discussing the paper with the editorial team and an academic editor with relevant expertise, I'm pleased to invite you to revise the paper in response to the reviewers' comments. Please note that we plan to send the revised paper to some or all of the original reviewers, and we cannot provide any guarantees at this stage regarding publication.

We ask that you submit your revision by June 19th. However, if this deadline is not feasible, please contact me by email, and we can discuss a suitable alternative.

Don't hesitate to contact the handling editor with any questions (sbruijn@plos.org).

Kind regards,

Heather

Heather Van Epps, PhD

Consulting Editor

[on behalf of]

Suzanne De Bruijn, PhD

Associate Editor

PLOS Medicine

sbruijn@plos.org

Comments from the reviewers:

Reviewer #1 (Statistics):

This is an obviously important topic, and the logic of the study is robust and the reporting is detailed. While this is based on observational data, it is possible to be the most reliable data, given that relevant trial data is not and probably will not be available.

However, I have several comments, some of which might be important to the robustness of the findings:

1. In the models that estimate the coverage decline and seroconversion by age, it seems that the primary predictors (e.g. week/age) were all modelled assuming linear relationship. While the model fit with data points on main manuscript p6 appear ok, it'd be good to check for nonlinearity by e.g. fitting fractional polynomials or splines to ensure these won't affect the conclusions.

2. Given the observational nature and the various assumptions behind the models, I wondered if the interpretation is too strong. I'd suggest to at least adding 'under current estimation' before the interpretations in the Abstract.

3. Similarly, won't it be a limitation that the estimates were based on observational data and confounding is possible.

Reviewer #2:

This is a very interesting paper that used a modelling approach to estimate poliovirus type 2 (PV2) immunity in under-5 children in 114 countries using bOPV in routine immunization. They conducted a systematic review of studies on PV2 seroconversion by age of IPV administration (with one or two doses and with full and fractional doses); then used a binomial regression model to fit seroconversion as a function of age at the time of the first dose and the time interval between doses; finally, considering also countries' data on routine immunization coverage as a function of age and data on doses delays, they estimated immunity of under-5 children to PV2 in each country, in 2025 and 2030. The results demonstrate median PV2 immunity of only 60% in 2025, that would increase to only 66% in 2030, if the vaccination schedule of the country is kept the same (with 19 countries with PV2 immunity <50% in 2030). The results point out that, besides the number of IPV doses and vaccine coverage, the age of IPV administration is important to achieve better PV2 immunity. The paper results may help public health managers to choose the best schedule to achieve high population immunity to PV2.

The study was well conducted, and the paper is well written. I did not have any suggestions to improve it

Reviewer #3:

The approach taken is attractive in its data driven nature; although there are many data challenges, the authors have taken the time to organise what data there are to maximise its usefulness - and the combination of a simple/robust but still mathematically useful approach to estimating immunity levels.

Reviewer #4:

The model really is very simple. It is little more than multiplying the country-specific coverage by the efficacy. This does mean that there is little to go wrong, though. My only real criticisms are these:

1) There is no attempt to check (or validate) the model. Many serological surveys have been undertaken over the years, and some comparison of these to the estimated seroprevalence, should really be done. A much weaker, but also interesting check would be to compare the risk of a cVDPV outbreak with the estimated coverage.

2) bOPV use is not taken into account. Hundreds of millions (even billions?) of doses of mOPV2 and nOPV2 have been used since 2016. The effect of IPV campaigns is taken into account. I don't understand why OPV campaigns are not also taken into account.

3) Clarke and Sanderson (Lancet 2009) should be referenced, when taking into account delays to vaccination. The method you use is based on this. There are also other papers that predate the reference that you do give (e.g. doi: 10.1136/jech.2010.124651.)

Reviewer #5:

The study would be strengthened with confidence intervals or probabilistic sensitivity analyses around projected immunity estimates.

Ignoring serotype-2 immunity from outbreak response campaigns underestimates total population immunity and risks.

The text is dense in some sections. Figures are highly detailed but may overwhelm readers; consider supplemental simplification or policy briefs.

---

(Note: not all will apply to your paper, but please check each item carefully and include your reponses as part of your point-by-point, noting N/A where appropriate)

* Please upload any figures associated with your paper as individual TIF or EPS files with 300dpi resolution at resubmission; please read our figure guidelines for more information on our requirements: http://journals.plos.org/plosmedicine/s/figures. While revising your submission, please upload your figure files to the PACE digital diagnostic tool, https://pacev2.apexcovantage.com/. PACE helps ensure that figures meet PLOS requirements. To use PACE, you must first register as a user. Then, login and navigate to the UPLOAD tab, where you will find detailed instructions on how to use the tool. If you encounter any issues or have any questions when using PACE, please email us at PLOSMedicine@plos.org.

* Please ensure that the study is reported according to the STROBE guideline (or appropriate extension) and include the completed STROBE checklist as Supporting Information. When completing the checklist, please use section and paragraph numbers, rather than page numbers. Please add the following statement, or similar, to the Methods: "This study is reported as per STROBE guideline (S1 Checklist).”

FIGURES AND TABLES

* Please consider avoiding the use of red and green in order to make your figure more accessible to those with colour blindness.

SUPPLEMENTARY MATERIAL

REFERENCES

OBSERVATIONAL STUDIES

* Abstract: Please include the study design, population and setting, number of participants, years during which the study took place (enrollment and follow up), length of follow up, and main outcome measures.

* Please ensure that the study is reported according to the STROBE (or appropriate STOBE extension; eg, RECORD) guideline (available from: https://www.equator-network.org/reporting-guidelines/strobe) and include the completed STROBE (or STROBE extension) checklist as Supporting Information. Please add the following statement, or similar, to the Methods: "This study is reported as per the Strengthening the Reporting of Observational Studies in Epidemiology (STROBE) guideline (S1 Checklist)." When completing the checklist, please use section and paragraph numbers, rather than page numbers.

* For all observational studies, in the manuscript text, please indicate: (1) the specific hypotheses you intended to test, (2) the analytical methods by which you planned to test them, (3) the analyses you actually performed, and (4) when reported analyses differ from those that were planned, transparent explanations for differences that affect the reliability of the study's results. If a reported analysis was performed based on an interesting but unanticipated pattern in the data, please be clear that the analysis was data driven.

* Please state in the Methods section whether the study had a prospective protocol or analysis plan. If a prospective analysis plan (from your funding proposal, IRB or other ethics committee submission, study protocol, or other planning document written before analyzing the data) was used in designing the study, please include the relevant document(s) with your revised manuscript as a Supporting Information file to be published alongside your study and cite it in the Methods section. A legend for this file should be included at the end of your manuscript. If no such document exists, please make sure that the Methods section transparently describes when analyses were planned, and when/why any data-driven changes to analyses took place. Changes in the analysis, including those made in response to peer review comments, should be identified as such in the Methods section of the paper, with rationale.

MODELLING STUDIES

The following list is derived from Geoffrey P Garnett, Simon Cousens, Timothy B Hallett, Richard Steketee, Neff Walker. Mathematical models in the evaluation of health programmes. (2011) Lancet DOI:10.1016/S0140-6736(10)61505-X:

* If pertinent, please provide a diagram that shows the model structure, including how the natural history of the disease is represented, the process and determinants of disease acquisition, and how the putative intervention could affect the system.

* Please provide a complete list of model parameters, including clear and precise descriptions of the meaning of each parameter, together with the values or ranges for each, with justification or the primary source cited and important caveats about the use of these values noted.

* Please provide a clear statement about how the model was fitted to the data, including goodness-of-fit measure, the numerical algorithm used, which parameter varied, constraints imposed on parameter values, and starting conditions.

* For uncertainty analyses, please state the sources of uncertainties quantified and not quantified [can include parameter, data, and model structure].

* Please provide sensitivity analyses to identify which parameter values are most important in the model. Uncertainty estimates seek to derive a range of credible results on the basis of an exploration of the range of reasonable parameter values. The choice of method should be presented and justified.

* Please discuss the scientific rationale for the choice of model structure and identify points where this choice could influence conclusions drawn. Please also describe the strength of the scientific basis underlying the key model assumptions.

---

## [Decision Letter · Decision Letter 2]

8 Sep 2025

Dear Dr. Gray,

Thank you very much for re-submitting your manuscript "Estimating population immunity against serotype-two poliomyelitis from the inactivated polio vaccine in routine immunisation across 112 countries" (PMEDICINE-D-25-00561R2) for review by PLOS Medicine. Please accept my apology in the delay in providing you with a decision, which was due to attempting to obtain reviewer advice.

I have discussed the paper with my colleagues and the academic editor and it was also seen again by 1 reviewer. I am pleased to say that provided the remaining editorial and production issues are dealt with we are planning to accept the paper for publication in the journal.

We look forward to receiving the revised manuscript by Sep 15 2025 11:59PM.

Sincerely,

Suzanne De Bruijn, PhD

Associate Editor

PLOS Medicine

plosmedicine.org

Requests from Editors:

GENERAL EDITORIAL REQUESTS

* Thank you for providing an Author Summary; Ideally each sub-heading should contain 2-3 single sentence, concise bullet points containing the most salient points from your study. In the final bullet point of ‘What Do These Findings Mean?’ Please include the main limitations of the study in non-technical language.

Please see our author guidelines for more information: "https://journals.plos.org/plosmedicine/s/revising-your-manuscript#loc-author-summary."

* Please confirm that your title complies with to PLOS Medicine's style. Your title must be nondeclarative and not a question. It should begin with main concept if possible. "Effect of" should be used only if causality can be inferred, i.e., for an RCT. Please place the study design ("A randomized controlled trial," "A retrospective study," "A modelling study," etc.) in the subtitle (ie, after a colon).

* Please confirm that your abstract complies with our requirements, including format (three sections: Background, Methods and Findings, and Conclusions) and providing all the information relevant to this study type https://journals.plos.org/plosmedicine/s/submission-guidelines#loc-abstract

* Please confirm that all numbers presented in the abstract are present and identical to numbers presented in the main manuscript text.

GENERAL

* In the author summary, in the final bullet point of 'What Do These Findings Mean?', please include the main limitations of the study in non-technical language.

* Currently the majority of the Methods are in the supplements. Please provide more details (if not all) on the Methods in the main text for transparency. As a reminder, we don’t have a word-limit.

FUNDING STATEMENT

* The funding statement should include: specific grant numbers, initials of authors who received each award, URLs to sponsors’ websites. Also, please state whether any sponsors or funders (other than the named authors) played any role in study design, data collection and analysis, the decision to publish, or preparation of the manuscript. If they had no role in the research, include this sentence: “The funders had no role in study design, data collection and analysis, decision to publish, or preparation of the manuscript.”

DATA AVAILABILITY

* Thank you for stating that the data is freely available, as well as providing the Github URL.

* Because Github depositions can be readily changed or deleted, we encourage you to make a permanent DOI'd copy (e.g. in Zenodo) and provide the URL.

* Please include all-data sources in your data availability statement.

* Please include the statement on code availability in the data availability statement.

* Regarding the seroconversion study data, please provide these data in a more easily accessible format. Are these data provided in Table 24 in the supplementary materials? If so, please specify this explicitly in the Data Availability Statement. If not, please provide a table with all the data used in this study, as well as the sources these data came from.

FIGURES

* Please define all elements of box plots in the figure caption - center line, box limits and whiskers.

* Please convert any stacked bar charts to another data representation for example a table, or other type of graph.

* Please consider avoiding the use of red and green in order to make your figure more accessible

* Please confirm that the appropriate usage rights apply to the use of this map. Please see our guidelines for map images: https://journals.plos.org/plosmedicine/s/figures#loc-maps

Figure 1: please label the subpanels, so it’s clear for everyone which subfigure is referred to.

Comments from Reviewers:

Reviewer #1: Thank you for addressing my comment. I think my last point on confounding has not come through well enough - I do not doubt the data source to be very good. What I was not certain is the projected immunity from the national data. More details in the limitations around this, i.e. the projection inaccuracy, would be welcomed.

---

## [Editor Report · Decision Letter 3]

7 Nov 2025

Dear Dr. Gray,

Thank you very much for re-submitting your manuscript "Estimating population immunity against serotype-two poliomyelitis from the inactivated polio vaccine in routine immunisation across 112 countries" (PMEDICINE-D-25-00561R3) for review by PLOS Medicine.

Thank you for addressing our requests. However, there are several remaining requests, which you can find at the bottom of this email. Please address these remaining concerns and provide us with a point-by-point response. Please also don't hesitate to contact me if anything is unclear, or if you have any other questions.

We look forward to receiving the revised manuscript by Nov 14 2025 11:59PM.

Sincerely,

Suzanne De Bruijn, PhD

Associate Editor

PLOS Medicine

plosmedicine.org

Requests from Editors:

* Please change your title to: “estimating population immunity against serotype-two poliomyelitis from the inactivated polio vaccine in routine immunisation across 112 countries: a modelling study”.

* Abstract, line 30. It mentions ‘whole county’, do you mean ‘whole country’? (same in the author summary, line 64).

* Author summary: Please change your subheadings to ‘why was this study done’, ‘what did the researchers do and find’ and ‘what do these findings mean’.

*Author Summary: under ‘what did the researchers do and find, please change the text to the following bullet points:

* We developed a model to estimate national population immunity against type-2 poliomyelitis across 112 countries in children < 5 years old.

* We projected immunity levels for current immunisation schedules, alternative scenarios, and catch-up vaccination campaigns.

* Current IPV-induced immunity levels are critically low in many locations, with 25 countries or territories having <50% population immunity in children under five years of age.

* We found that the IPV schedules and coverage are suboptimal in many countries, and in many cases adopting the 14-39 week schedule would be beneficial.

* Author Summary: under ‘what do these findings mean’ please change the text to:

* The 40 countries with an early immunisation schedule, and the 23 countries using a single dose of IPV immunisation should all change to a 14-39 week schedule.

* Catch-up campaigns could be considered to shorten the time to reach higher immunity in countries with recent introductions of a second dose of IPV, and in countries where IPV coverage is very low and immunity would otherwise remain critically low.

* The work is limited by the fact that IPV provides only a partial picture of total immunity where there has been emergency type-two live vaccine use. Similarly, the use of whole county estimated may mask subnational coverage differences and pockets of extremely low immunity.

* Financial disclosure: Please include a URL to the funders websites.

* Financial disclosure: it appears 2 of the study authors are affiliated with one or more of the agencies that funded the study. Please include the following statement in your financial disclosure: "[Author name] is [author's role] at [funding agency]. The funders had no other role in study design, data collection and analysis, decision to publish, or preparation of the manuscript.”

Please also make these changes in the metadata in Editorial Manager.

* Thank you for providing your Github URL. However, there seems to be a mistake, as the URL is not working. Please double check and provide us with the correct link.

* Please make a permanent DOI’d copy of your github deposition (e.g. in Zenodo) and provide us with the URL. I appreciate you state that this will be completed before publication, but we need to include the URL in the manuscript. If there is a reason this is delayed, please let me know and we can discuss the options.

* If you are providing the used code on your github page (or Zenodo-created URL), please include the following text in your data availability statement: “Custom code used in this study is available at [URL]”.

*Thank you for including the seroconversion studies in a .CSV file. Please ensure that the link to the repository is correct. Please also include the following text in your data availability statement: “seroconversion study data can be found in [file] at [URL].

* Please provide the data underlying the stacked bargraph in figure 1A as a supplementary table, and reference this supplementary table in the legend.

* Figure 5A: Please change this one back to the original stacked bar graphs, as these were more informative. Please provide the underlying data as a supplementary table, and reference this supplementary table in the legend.

---

## [Editor Report · Decision Letter 4]

9 Feb 2026

Dear Dr Blake,

On behalf of my colleagues and the Academic Editor, James Beeson, I am pleased to inform you that we have agreed to publish your manuscript "Estimating population immunity against serotype-two poliomyelitis

from the inactivated polio vaccine in routine immunisation across 112 countries: a modelling study" (PMEDICINE-D-25-00561R4) in PLOS Medicine.

We have a few remaining requests:

1) Please define all acronyms used in each figure or table in its corresponding legend.

2) Please consider avoiding the use of red and green in order to make your figure more accessible (Figure 5A and supplement Figure 1).

3)Supplements: table 2.1 doesn’t have a header: please amend.

4) Thank you for providing the data underlying figure 1: please change the text from “see data table in Supplementary Materials section 6” to ‘data underlying figure 1A can be found in supplementary materials section 6, table 25.

5) Change the text from “See Supplementary Material Section 2 for data tables” to ‘data underlying figure 5A can be found in supplementary material section 2. Also, I appreciate that all the data can be found in this section, but please provide a table with the actual numbers that can be found in this figure.

Furthermore, before your manuscript can be formally accepted you will need to complete some formatting changes, which you will receive in a follow up email. Please be aware that it may take several days for you to receive this email; during this time no action is required by you. Once you have received these formatting requests, please note that your manuscript will not be scheduled for publication until you have made the required changes.

PRESS

Sincerely,

Suzanne De Bruijn, PhD

Associate Editor

PLOS Medicine